# Cryo-electron tomography reveals novel features of a viral RNA replication compartment

Kenneth J Ertel[1,2†‡], Desirée Benefield[1,3†], Daniel Castaño-Diez[4], Janice G Pennington[1,2], Mark Horswill[1,3], Johan A den Boon[1,3], Marisa S Otegui[5,6], Paul Ahlquist[1,2,3*]

[1]Institute for Molecular Virology, University of Wisconsin-Madison, Madison, United States; [2]Howard Hughes Medical Institute, University of Wisconsin-Madison, Madison, United States; [3]Morgridge Institute for Research, University of Wisconsin-Madison, Madison, United States; [4]BioEM lab, Biozentrum, University of Basel, Basel, Switzerland; [5]Department of Botany, University of Wisconsin-Madison, Madison, United States; [6]Laboratory of Cell and Molecular Biology, University of Wisconsin-Madison, Madison, United States

*For correspondence: ahlquist@wisc.edu

[†]These authors contributed equally to this work

Present address: [‡]Janelia Research Campus, Howard Hughes Medical Institute, Ashburn, United States

Competing interests: The authors declare that no competing interests exist.

**Abstract** Positive-strand RNA viruses, the largest genetic class of viruses, include numerous important pathogens such as Zika virus. These viruses replicate their RNA genomes in novel, membrane-bounded mini-organelles, but the organization of viral proteins and RNAs in these compartments has been largely unknown. We used cryo-electron tomography to reveal many previously unrecognized features of Flock house nodavirus (FHV) RNA replication compartments. These spherular invaginations of outer mitochondrial membranes are packed with electron-dense RNA fibrils and their volumes are closely correlated with RNA replication template length. Each spherule's necked aperture is crowned by a striking cupped ring structure containing multifunctional FHV RNA replication protein A. Subtomogram averaging of these crowns revealed twelve-fold symmetry, concentric flanking protrusions, and a central electron density. Many crowns were associated with long cytoplasmic fibrils, likely to be exported progeny RNA. These results provide new mechanistic insights into positive-strand RNA virus replication compartment structure, assembly, function and control.

## Introduction

Positive-strand RNA viruses, whose virions contain mRNA-sense single stranded RNA, represent the largest genetic class of viruses (*King et al., 2012*), including many human, animal, and plant pathogens. These viruses complete their entire replication cycle in the host cell cytoplasm. To generate sites for their genomic RNA replication, positive-strand RNA viruses expand and rearrange host cellular membranes (*Fernández de Castro et al., 2017*; *den Boon et al., 2010*; *Kovalev et al., 2016*; *Paul and Bartenschlager, 2013*; *Paul et al., 2014*; *Romero-Brey and Bartenschlager, 2014*). Such membrane-associated RNA replication sites are thought to concentrate essential viral and host factors, and to organize successive steps for efficient viral RNA synthesis (*Atasheva et al., 2010*; *Neuvonen et al., 2011*). Depending on their architecture and topology, such sites also might provide some protection from cellular nucleases and recognition by antiviral double-stranded RNA (dsRNA) sensors, such as RIG-I, MDA-5, and AGO (*Ahmad and Hur, 2015*; *Andrejeva et al., 2004*; *Diao et al., 2007*; *Fagard et al., 2000*; *van Rij et al., 2006*; *Weinmann et al., 2009*; *Yoneyama et al., 2004*) and help to spatially segregate the viral RNA genome's competing

functions as an mRNA for viral protein synthesis, a template for viral RNA synthesis, and a substrate for encapsidation. RNA replication on expanded, rearranged cellular membranes is universal among positive-strand RNA viruses and thus presents a target for broadly applicable antiviral therapies.

The structural organization of viral RNAs and proteins at such membrane-associated sites of RNA replication has been largely unknown. For example, in various studies, viral RNA replication templates and replication proteins have contrastingly been suggested to reside on the cytoplasmic face of membranes or inside vesicular compartments (*Belov et al., 2007*, *2012*; *Diaz et al., 2012*, *2010*; *Knoops et al., 2012*; *Kopek et al., 2007*; *Miller et al., 2001*; *Schwartz et al., 2002*; *Welsch et al., 2009*).

To address these questions, we turned to flock house virus (FHV), a well-studied, model positive-strand RNA virus in the nodavirus family. The FHV genome organization is shown in *Figure 1A*. The bi-partite genome comprises RNA1 and RNA2, encoding the viral replicase protein A and the viral capsid proteins, respectively (*Venter and Schneemann, 2008*). RNA1 further directs the synthesis of RNA3, a subgenomic mRNA encoding protein B2, which counters host RNA interference defenses (*Chao et al., 2005*; *Li et al., 2002*; *Lingel et al., 2005*). RNA3 retains part of the protein A coding sequence and thus potentially encodes an uncharacterized C-terminal protein A segment, referred

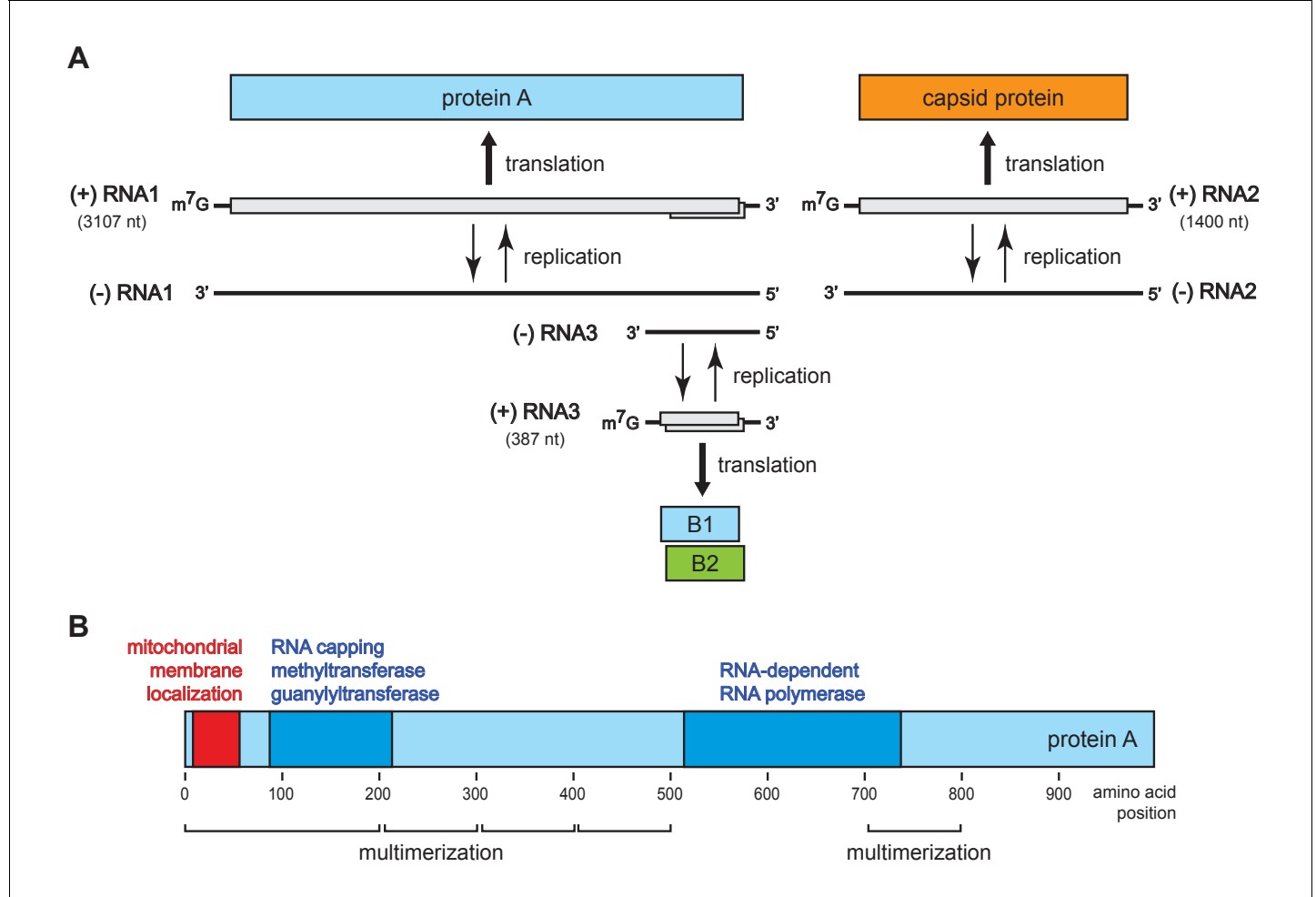

**Figure 1.** Flock house virus genome organization and functional map of replicase protein A. (A) The bipartite FHV RNA genome encodes the viral replicase protein A on RNA1 and the capsid protein precursor on RNA2. Protein A mediates synthesis of a negative strand copy of the genomic RNAs which serves as template for synthesis of progeny positive strand RNAs, and an additional partial negative strand copy of RNA1 to template a 3' co-terminal subgenomic RNA3. RNA3 potentially encodes protein B1, co-terminal with the C-terminus of protein A, and protein B2 which has anti-RNAi function. (B) A linear representation of protein A annotated to show membrane-association and enzymatic domains, and several regions independently capable of homotypic multimerization.

to as B1. *Figure 1B* shows the organization of protein A functional domains, including an N-terminal mitochondrial membrane association domain (*Miller and Ahlquist, 2002*), an RNA-dependent RNA polymerase domain, an RNA capping domain, and multiple independent multimerization regions identified by fluorescence resonance energy transfer (*Dye et al., 2005b*).

Previous transmission electron microscopy (TEM) and EM tomography by our group and others showed that FHV infection of *Drosophila* cells induces ~50 nm diameter 'spherule' invaginations of the outer mitochondrial membranes (*Kopek et al., 2007*; *Lanman et al., 2008*; *Miller et al., 2001*; *Short et al., 2016*). These spherules are the sites of viral protein A replicase accumulation and viral RNA synthesis (*Kopek et al., 2007*), and remain connected to the cytoplasm via a neck-like aperture (*Kopek et al., 2007*; *Miller et al., 2001*). Similar spherule RNA replication compartments are formed on various intracellular membranes by many positive-strand RNA viruses, such as the human-infecting flaviviruses, alphaviruses, and many others (*Belov et al., 2007*, *2012*; *den Boon and Ahlquist, 2010*; *den Boon et al., 2010*; *Diaz et al., 2012*, *2010*; *Knoops et al., 2012*; *Kopek et al., 2007*; *Miller et al., 2001*; *Schwartz et al., 2002*; *Welsch et al., 2009*). While informative, these older images were derived from FHV-infected and other virus-infected cells and cell fractions that were chemically fixed and embedded in plastic, which are prone to fixation-induced artifacts (*McDonald and Auer, 2006*). Moreover, these heavy metal stained samples defined the low-resolution ultrastructure of the bounding spherule membranes, but did not allow visualizing the viral RNAs and replication proteins.

Here we report visualization of FHV RNA replication compartments in mitochondria from FHV-infected *Drosophila* cells by cryo-electron microscope (cryo-EM) tomography. Cryo-EM presents samples in a much more native state, bypasses heavy metal staining, and images the sample's intrinsic electron density, providing direct visualization of all components – lipid, RNA, and protein – of the RNA replication complexes (*Bertin et al., 2012*; *Gold et al., 2014*; *Meyerson et al., 2011*). Moreover, we used sub-tomogram averaging to powerfully enhance image resolution. The resulting images reveal numerous striking new features, including previously unrecognized features of the spherule membrane structure, crowning of the cytoplasmic side of the spherule neck by a dramatic 12-fold symmetrical structure containing FHV replicase protein A, and densely coiled interior filaments and single exterior filaments strongly implicated as viral template and product RNAs. Together with complementary genetic manipulations and biochemical results, the data provide substantial insights not only into replication complex structure but also function and assembly.

## Results

### Cryo-EM reveals new interior and exterior features of FHV RNA replication compartments

Previously our group showed that mitochondria isolated from FHV-infected *Drosophila* cells retain RNA replication compartments (spherules) that are active in viral RNA synthesis with a specific activity approaching that in infected cells (*Kopek et al., 2007*). Accordingly, to image FHV RNA replication compartments in a near native state while avoiding distortions and artifacts due to chemical fixation (*McDonald and Auer, 2006*), we isolated mitochondria from mock-infected and FHV-infected *Drosophila* cells and arrested them by rapid plunge freezing. These vitrified, hydrated, unstained samples then were examined by cryo-EM using a high resolution direct electron detector. As expected, mitochondria from mock-infected cells had closely appressed outer and inner mitochondrial membranes, with inner membrane cristae extensions into the interior matrix (*Figure 2A*). By contrast, and consistent with previous reports (*Kopek et al., 2007*, *2010*; *Lanman et al., 2008*; *Miller et al., 2001*; *Short et al., 2016*), the outer and inner membranes of mitochondria from FHV-infected cells were separated, and the intervening space filled with numerous membrane spherules. While, in chemically-fixed samples, FHV RNA replication complexes had oblong shapes significantly elongated perpendicular to the outer mitochondrial membrane (*Kopek et al., 2007*; *Miller et al., 2001*), in plunge-frozen samples they were much more spherical (*Figure 2B*). As in our prior studies (*Kopek et al., 2007*), we confirmed that mitochondria isolated from FHV-infected cells retained the biological integrity of the spherules that were highly active in viral RNA synthesis (*Figure 2C*), and preserved all known features of spherule architecture as seen in infected whole cells.

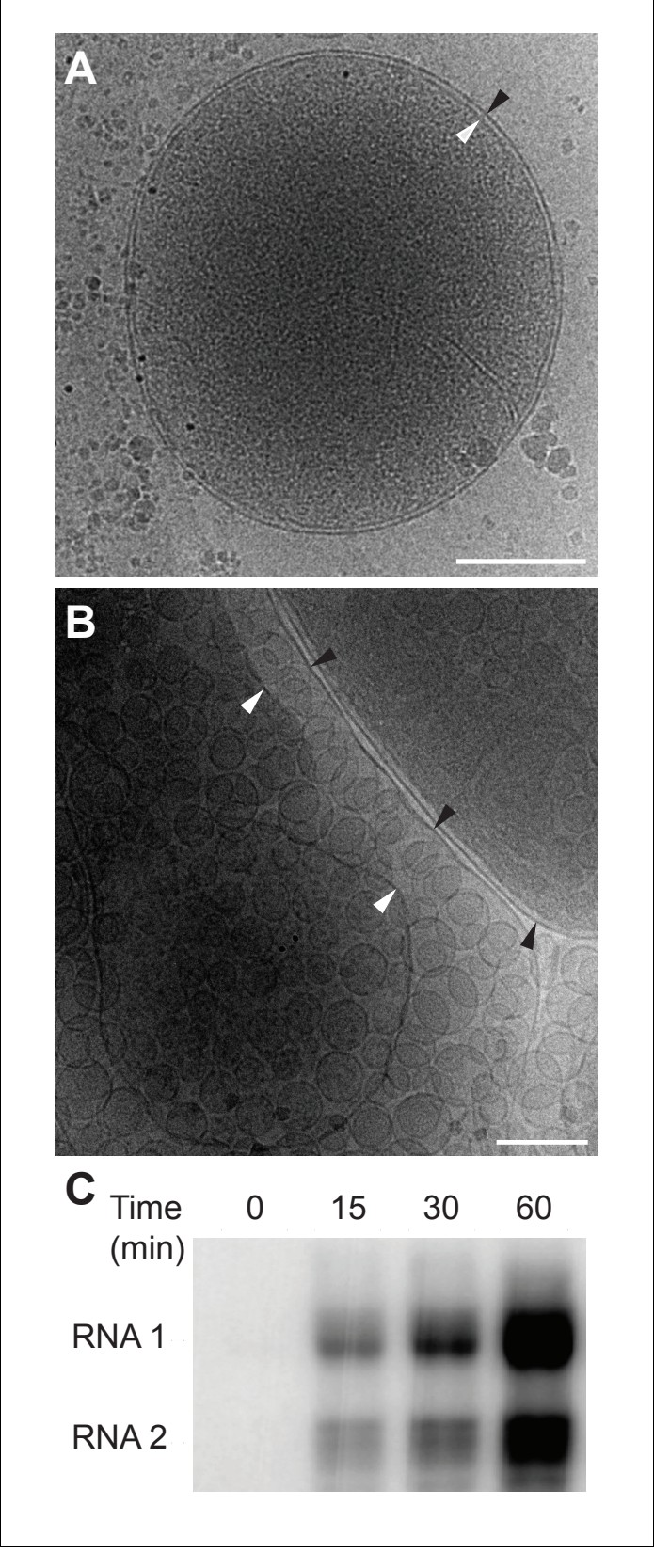

**Figure 2.** Cryo-EM images of Drosophila S2 cell mitochondria. Mitochondria were isolated from mock-infected or FHV-infected S2 cells, applied to carbon-coated grids, plunge-frozen, and imaged using a TF-30 electron
*Figure 2 continued on next page*

*Figure 2 continued*

microscope under cryogenic conditions. (**A**) Intact mitochondrion with inner (white arrowhead) and outer (black arrowhead) membranes approximately 10 nm apart. Scale bar = 200 nm. (**B**) Mitochondria isolated from FHV-infected cells were closely associated with numerous round vesicular compartments in the extensively dilated lumen between inner and outer membranes. Scale bar = 100 nm. (**C**) Mitochondria preparations from FHV-infected cells show active continued viral RNA synthesis when incubated in the presence of radiolabeled [32P]UTP. Results are representative of three independent experiments.

To reveal additional critical features and structural relationships, we acquired cryo-electron tomography (cryo-ET) tilt series of images of mitochondria from FHV-infected and uninfected cells and processed these to reconstruct three-dimensional views. *Video 1* shows such a tomographic reconstruction of a typical mitochondrion from an FHV-infected cell, including hundreds of spherule RNA replication compartments. As expected, rapid cryo-fixation improved preservation and direct imaging of the native electron density of these samples greatly facilitated visualizing non-membrane components of these replication compartments, including RNAs and protein structures, at much higher resolution and revealing multiple striking features. These included a wider range of spherule diameters than previously observed, coiled interior filaments filling the invaginated membrane vesicle, an ~25–30 nm diameter by 20 nm-high cupped density crowning the cytoplasmic side of each spherule neck, and usually single exterior filaments extending from these crown structures into the extra-mitochondrial cytoplasmic space (*Videos 1* and *2*). Below we present further imaging and analysis of the interior filaments, the relation of spherule volume to RNA template length, and the crown structure. The exterior filaments proceeding from crowns into the cytoplasm are considered further in the Discussion.

## Coiled interior filaments fill FHV RNA replication vesicles

Previous EM of chemically fixed, heavy metal-stained FHV spherules (*Kopek et al., 2007*, *2010*; *Lanman et al., 2008*; *Miller et al., 2001*; *Sosinsky et al., 2008*) and other positive-strand

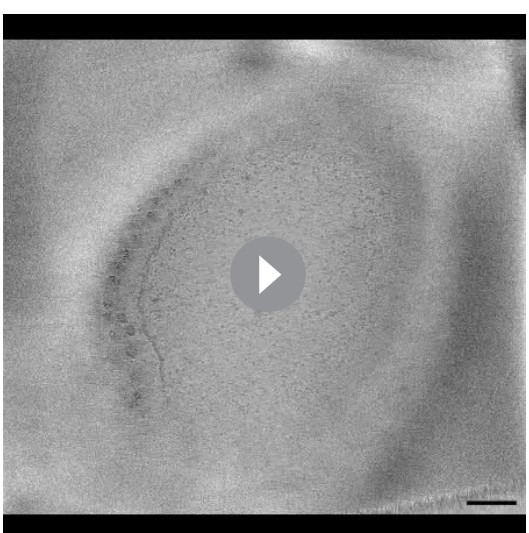

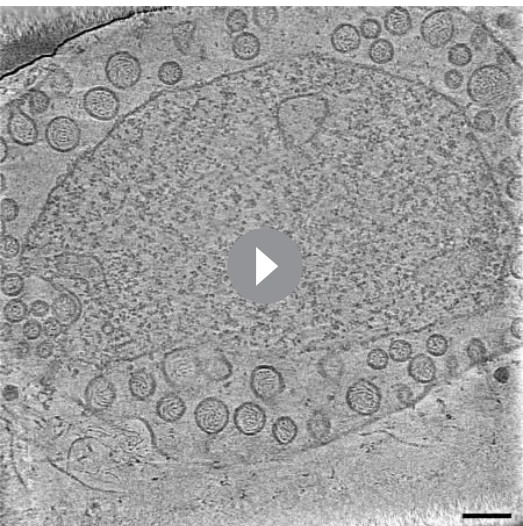

**Video 1.** Cryo-EM Tomogram of a Mitochondrion from FHV-infected cells. Complete tomographic reconstruction of a mitochondrion bearing numerous FHV spherules, at 19,500x magnification, and represented as a series of Z-stack images. Scale bar = 200 nm.

**Video 2.** Higher Magnification View of a Mitochondrion from FHV-infected cells. Tomographic reconstruction of the mitochondrion with FHV spherules shown in *Figure 3*. Note particularly the closely packed fibrillar material in the spherule interiors. At 41,000x magnification, represented as a series of Z-stack images. Scale bar = 100 nm.

RNA virus replication compartments (*Belov, 2016*; *Belov et al., 2012*; *Diaz et al., 2010*; *Junjhon et al., 2014*; *Knoops et al., 2012*; *Oudshoorn et al., 2016*) defined their membrane ultra-structure but, while showing some electron density inside such compartments, did not reveal any significant, consistent interior structure. Nevertheless, diverse biochemical and immunogold localization results from our group and others have shown that the replication compartments of FHV and many other positive-strand RNA viruses contain membrane-protected positive- and negative-strand template RNAs, likely in dsRNA form, and nascent progeny viral RNA (*Hatta and Francki, 1981*; *Kopek et al., 2007*; *Lee et al., 1994*; *Russo et al., 1983*; *Schwartz et al., 2002*). In keeping with this, our new cryo-EM imaging revealed FHV spherule interiors to be densely packed with filaments that appeared in cross-section as closely packed rods and in side views as long strands curved within the spherules (*Figure 3A*). *Videos 2* and *3* provide continuous views through the complete volume of numerous spherules, showing that many of the filament segments visualized in individual cross-sections are connected between successive tomographic slices. Highlighting this interior electron density with a computed tomography software package (AMIRA, FEI) provided three-dimensional views, further illustrating that these curved filaments occupy essentially the full spherule interior in a pattern reminiscent of a ball of yarn (*Figure 3B and C*). While the current tomographic resolution did not allow unambiguous tracing of the full length of these filaments, individual sectioned views such as in *Figure 3D* frequently displayed continuous, relatively straight segments at least 40 nm long, consistent with estimates of ~60–64 nm for the persistence length of dsRNA, i.e. the length of dsRNA required to show significant bending (Abels, Moreno-Herrero, *Abels et al., 2005*).

## Spherule replication compartment volume correlates with RNA template length

Our earlier studies using chemical fixation discerned no clear relation between FHV RNA template length and spherule size (*Kopek et al., 2007*, *2010*). However, as noted above, our current studies using cryo-fixation preserved more spherules and revealed a much greater range of spherule sizes. To explore this distribution of spherule sizes more precisely, we measured the volume of 1592 spherules from four reconstructed tomograms derived from FHV-infected cells across four independent experiments. The average spherule diameter was ~50 nm, consistent with prior observations (*Kopek et al., 2007*, *2010*; *Miller et al., 2001*), but the range of spherule diameters varied from ~30 to nearly 70 nm. The corresponding volume range was between 13,600 $nm^3$ and 145,500 $nm^3$, with an average spherule volume of 66,100 $nm^3$ and a standard deviation of ±32,900 $nm^3$. When plotted, the distribution of spherule volumes showed three to four peaks, suggesting the possibility of distinct size classes of spherules (*Figure 4A*). All FHV spherules observed in these cryo-EM experiments contained interior filaments at similar packing density (*Figure 3* and *Video 3*), implying that spherules of varying diameter must contain varying amounts of RNA. Thus, the presence of multiple spherule size classes in full FHV infection might be related to the replication of three viral RNA templates, genomic RNA1 and RNA2, and subgenomic (sg) RNA3, with respective lengths of 3.1, 1.4, and 0.4 kb, occurring in different spherules.

To test such possible relationships between viral RNA templates and spherule size, spherule volumes were measured in mitochondria from *Drosophila* cells transiently expressing FHV RNA replication protein A (from a non-replicating mRNA) and a defined FHV RNA replication template transcribed in vivo from a DNA plasmid. *Figure 4B* shows the results for cells expressing protein A and an FHV genomic RNA1 template: volumes measured for 541 spherules from three tomograms from two independent transfection experiments revealed that this combination produced two distinct spherule size populations reflecting the extremes of the distribution produced by full FHV infection, consistent with the fact that RNA1 replication gives rise to 3.1 kb RNA1 and 0.4 kb sgRNA3 progeny. The larger spherule class distributed around a volume of ~108,000 $nm^3$, the smaller spherule class around ~12,000 $nm^3$. Similarly, *Figure 4C* shows the results for cells expressing protein A and an FHV sgRNA3 template: the resulting spherules volumes closely matching the smaller size class of spherules induced by simultaneous replication of RNA1 and RNA3. Thus, the size distribution of FHV RNA replication compartments is strongly correlated with the length of the viral RNA template or templates being replicated.

Subtracting the RNA1- and RNA3-associated spherule peaks from the size distribution for full infection leaves a broad range of intermediate sizes possibly containing more than the single peak that might be expected for replication of the remaining 1.4 kb genomic RNA2 (*Figure 4A*). RNA2

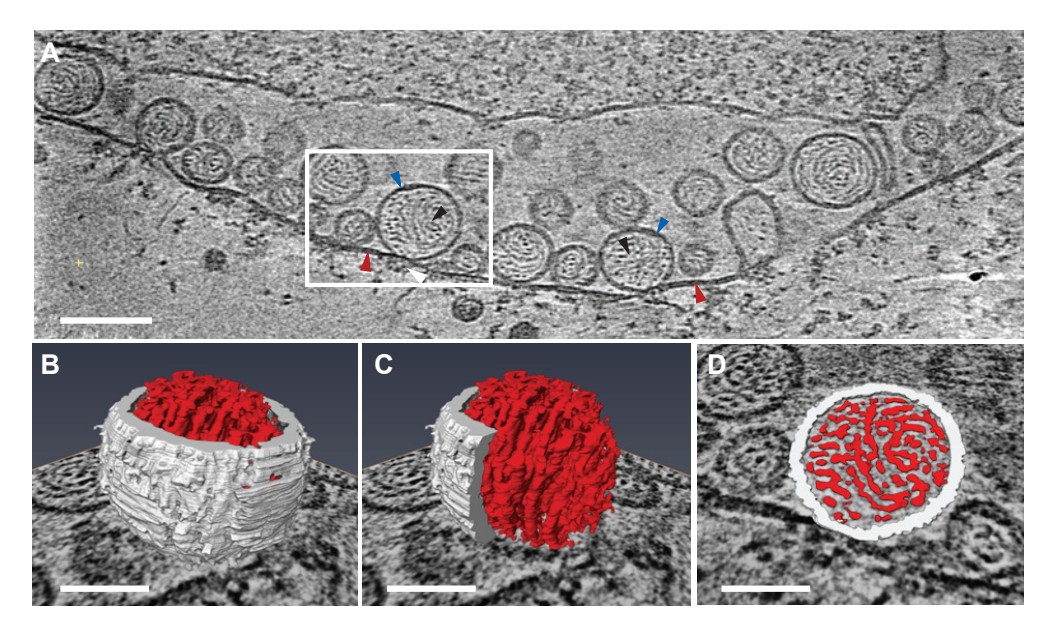

**Figure 3.** Viral RNA and protein complex association with FHV spherules. (**A**) Cryo-ET image of a mitochondrion associated with FHV spherules, taken from a reconstructed tomogram obtained at 41,000 x magnification, defocus of −6. Mitochondrial outer membrane, spherule membrane, interior spherule filaments, and spherule aperture structures are indicated with red, blue, black, and white arrowheads, respectively, consistent with arrowhead colors used in *Figures 5* and *6*. Scale bar is 100 nm. (**B–D**) Segmentation of the white-outlined area in panel A. The Amira program (FEI) was used to trace and structurally segment spherule membrane in white and interior spherule density in red. Panels B and C provide three dimensional renditions, panel D segments the structure in a single plane. Scale bars are 50 nm. See *Video 2* for complete tilt series.

The following figure supplement is available for figure 3:

**Figure supplement 1.** Distribution of center-to-center distance between interior spherule RNA filaments.

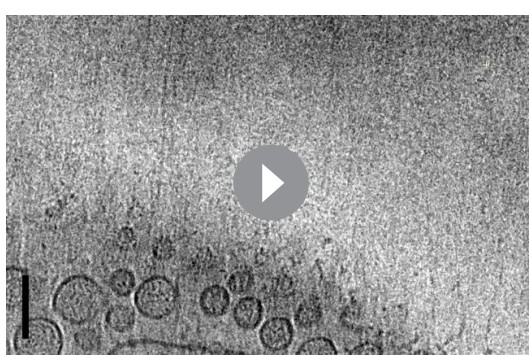

**Video 3.** Tomographic View of Crowns and Cytoplasmic Filaments on a Mitochondrion from FHV-infected cells. Detailed view of a portion the mitochondrion used in *Figure 5* and *Video 1*. Viral RNA exits FHV spherules in close proximity to the 'crown' protein complexes at the spherule necks. Scale bar = 100 nm.

replication requires trans-activation by RNA3 (*Lindenbach et al., 2002*; *Eckerle and Ball, 2002*), which could lead to some spherules containing both RNA2 and RNA3. Moreover, through ongoing processes of RNA synthesis and RNA export, the size and RNA content of individual spherules might vary dynamically. These and other issues that might contribute to the breadth of spherule size distributions associated with individual RNA templates (*Figure 4B and C*) and the apparent complexity of the size distribution for full infection (*Figure 4A*) are considered further in the Discussion.

To more precisely address the relationship between FHV RNA template length and spherule volume, we further considered packaging density under the assumption that the filaments in FHV spherules are dsRNA, which as noted above appears likely from biochemical and other data. The filaments and their arrangement in concentric, wound layers were generally similar

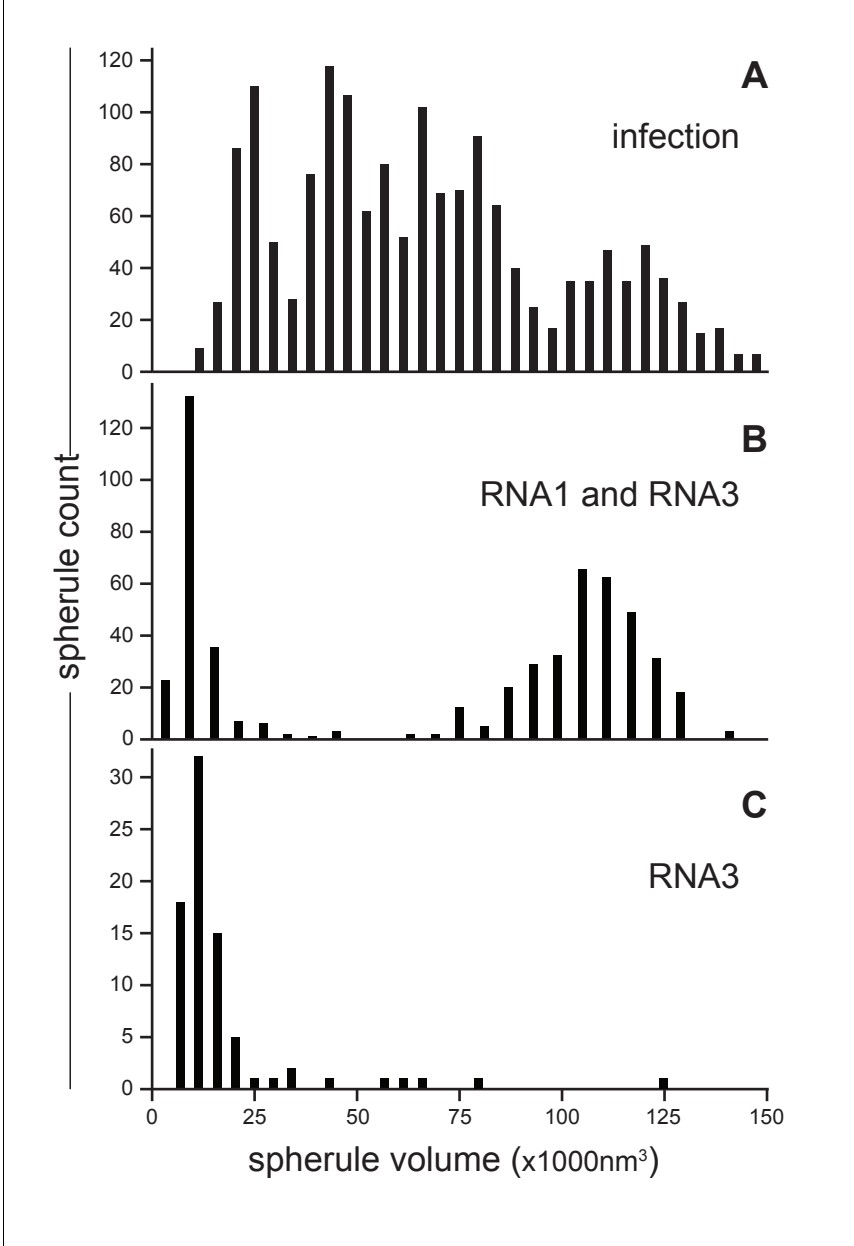

**Figure 4.** FHV RNA template length determines replication spherule vesicle volume. (**A**) FHV infection involves the simultaneous replication of RNA1, RNA2, and RNA3 resulting in a range of spherule volumes from 13,600 to 145,000 nm3. Spherule volumes were calculated from measurements of spherule radius at the widest circumference using IMOD software (*Nicastro et al., 2006*). (**B**) Cells that transiently express protein A from a non-replicating RNA in addition to RNA1 template have spherule volumes that sort into two pronounced populations suggesting a volume range for spherules containing either RNA1 or RNA3. (**C**) Cells expressing protein A and RNA3 show only the smaller spherule volume population.

to those of dsRNA templates in the replicative cores of dsRNA viruses (*Figure 3*, *Video 1*), and we used the published measurements on dsRNA in the replicative virion core particles of cytoplasmic polyhydrosis dsRNA virus (CPV) (*Zhang et al., 2015*) as a reference to calculate packaging density. Over short distances within which the RNA filaments of such dsRNA viruses or the FHV spherules are relatively straight, their packing density can be locally approximated as the packing of parallel cylinders, which is proportional to the square of their center-to-center distance. For CPV and FHV the average inter-filament distances are 2.7 nm (*Zhang et al., 2015*) and 6.4 nm (*Figure 3—figure*

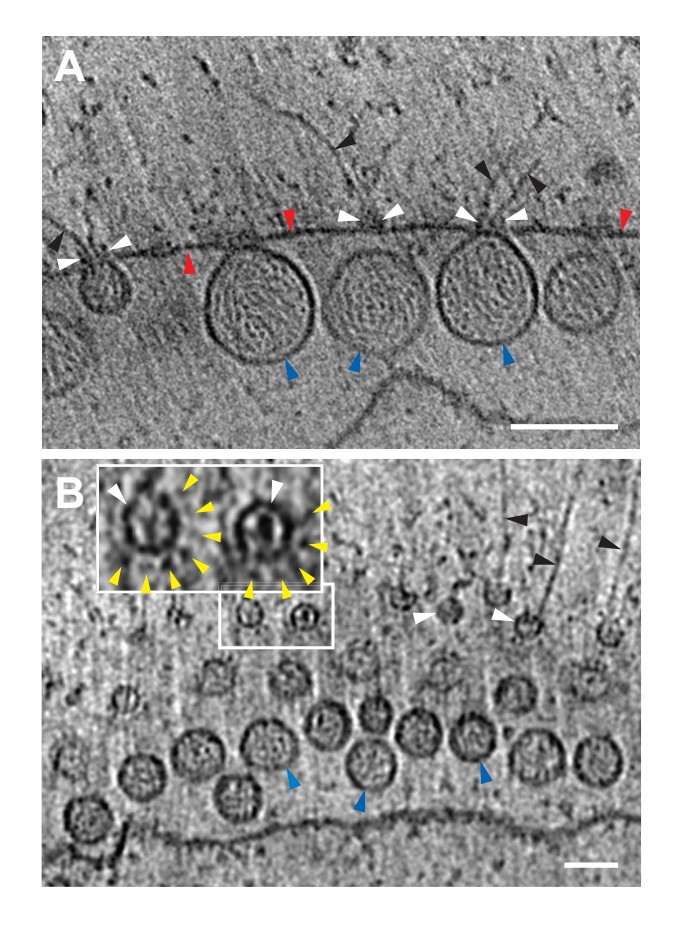

**Figure 5.** FHV spherule aperture densities are ordered protein complexes associates with viral RNA extrusions. (**A**) A single tomographic slice perpendicular to the outer membrane of a mitochondrion from an FHV-infected cell, showing multiple spherule cross sections. The image was obtained from a reconstructed tomogram at 19,500x magnification, defocus of −6. The protein complex at the spherule neck aperture is indicated with white arrowheads and extruding filaments (likely product RNA strands - see main text) with black arrowheads. The mitochondrial outer membrane and the spherule membrane are indicated with red and blue arrowheads, respectively. (**B**) Tomographic slice nearly parallel to the outer membrane of an FHV-modified mitochondrion. The upper portion shows multiple spherule apertures (two of which are boxed in white) viewed from above the cytoplasmic side of the membrane. Filamentous protusions associated with these apertures again are marked with black arrowheads. Inset: magnified view to show the aperture-surrounding concentric circular arrangement of additional smaller densities (yellow arrowheads). In the lower portion, the plane of sectioning passes below the outer mitochondrial outer membrane to section multiple spherule vesicles. Scale bars are 100 nm. Arrowhead colors are consistent with those used in *Figures 3* and *6*. Tomograms were gauss-filtered to remove noise. See *Videos 1* and *4* for complete tilt series and tomogram reconstructions.

The following figure supplement is available for figure 5:

**Figure supplement 1.** The 12-fold 'teeth' at the necks of FHV spherules.

*supplement 1*), respectively. Thus, the dsRNA packing density in FHV spherules should be ~5.8 fold lower than CPV's 334 bp / 1000 nm$^3$ (~24.7 kbp dsRNA genome per ~74,000 nm$^3$ [*Cheng et al., 2011*]). At a resulting calculated density of ~58 bp / 1000 nm$^3$, dsRNA1 (3.1 kbp), dsRNA2 (1.4 kbp), and dsRNA3 (0.4 kbp) would occupy ~53,000 nm$^3$, ~24,000 nm$^3$, and ~7000 nm$^3$, respectively. Accordingly, even the smallest of the RNA1-associated ~80,000–130,000 nm$^3$ spherules (*Figure 4B*) could accommodate a complete RNA1 dsRNA, consistent with the need for at least one negative-strand template to support positive-strand synthesis and our finding for each of FHV RNAs 1, 2 and

3 that similar levels of negative- and positive-strand fractionate from infected cells in a mitochondrially-associated, nuclease-protected state (N. Unchwaniwala and P. Ahlquist, unpublished results). Similarly, even the smallest of the RNA3-associated ~7,000–15,000 nm$^3$ spherules (*Figure 4B and C*) could accommodate a complete RNA3 dsRNA. Again, multiple mechanisms potentially contributing to a range of larger spherule sizes in these populations are noted above and considered further in the Discussion.

## FHV spherule necks are crowned by cupped structures

Our previous examination of chemically-fixed, plastic-embedded samples of FHV replication compartments in whole infected cells, using both two-dimensional transmission EM imaging (*Kopek et al., 2010*; *Miller et al., 2001*) and three-dimensional tomographic reconstruction (*Kopek et al., 2007*), had revealed the regular presence of electron-dense structures on the cytoplasmic surface of outer mitochondrial membranes immediately above the necks of FHV spherules. The cryo-EM images presented here revealed these neck-surmounting structures dramatically (*Figure 5A*, *Videos 2* and *3*). Cross-sectional views showed that these structures were cup-shaped and that their interiors were associated with extruding cytoplasmic filaments (*Figure 5A*). In top views perpendicular to the mitochondrial surface these structure appeared circular, with a small central electron density (*Figure 5B*). In many cases, close inspection of such top views revealed an outer concentric ring of regularly spaced small densities (*Figure 5B* inset, *Figure 5—figure supplement 1*).

To better visualize these striking new features, we performed iterative subtomogram averaging (*Castaño-Díez et al., 2012*) of the regions surrounding the apertures of over 300 individual spherules, and reconstructed a three-dimensional image with a resolution estimated at approximately 3.3 nm (*Figure 6G*, *Figure 6—figure supplement 1*, and *Video 4*). While no symmetry was artificially imposed, power spectrum calculations showed a strong peak at rotational Fourier component 12 (*Figure 6H*) and accordingly, the averaged reconstruction of the spherule aperture structure revealed a clear twelve-fold symmetry. The reconstructed structure showed a central ~19 nm diameter 'turret' projection, surrounded by an outer, ~35 nm diameter ring of twelve smaller 'teeth' (*Figure 6*) spaced ~4 nm apart, projected ~3 nm above the outer membrane surface, and angled slightly toward the turret (*Figure 6C–F*). This combined turret-and-teeth structure is hereafter referred to as a 'crown.' Contiguous groups of outer teeth were readily visible in the unaveraged top views of many individual crown subtomograms (*Figure 5B* and inset). However, due to the limited resolution of such individual cryo-tomograms prior to subtomogram averaging, it was not possible to clearly visualize this outer ring completely around individual crowns to determine whether each crown had a complete set of twelve teeth. Nevertheless, no gaps within an otherwise contiguous array of teeth were ever observed, implying that the occupation frequency across all 12 possible positions was high.

At its highest dimension, the turret was raised ~14 nm above the outer mitochondrial membrane surface and in cross-section was seen to reside in a slight depression of the outer mitochondrial membrane (*Figure 6A and E*). Within the turret, a central, ~11 nm diameter open channel into the spherule was in close agreement with previous size estimates of spherule neck apertures (*Kopek et al., 2007*; *Miller et al., 2001*; *Short et al., 2016*). A strongly averaged central density within the turret was consistent with the above-mentioned filamentous extrusions observed in the original tomograms (*Figure 5* and *Video 3*). Retention of this central density in the reconstructions implies that it must have been present in the majority of crown densities used for the subtomogram averaging, consistent with its regular visualization in top views of individual crowns (*Figure 5B*).

## FHV replication protein A is a major constituent of the spherule crown

The presence of crown structures on FHV spherules and their absence on mitochondria from mock-infected cells (*Video 5*) indicated that they either consist of or are induced by viral factors. From among FHV-encoded proteins (see Introduction), we considered the prime candidate to form crowns to be the 112 kDa, multifunctional FHV protein A, based on many prior observations: First, protein A, having RNA-dependent RNA polymerase, RNA capping, and other functions, is the sole FHV protein required for RNA replication (*Dye et al., 2005a*; *Miller and Ahlquist, 2002*; *Miller et al., 2003*). Protein A localizes to outer mitochondrial membranes via an N-terminal trans-membrane

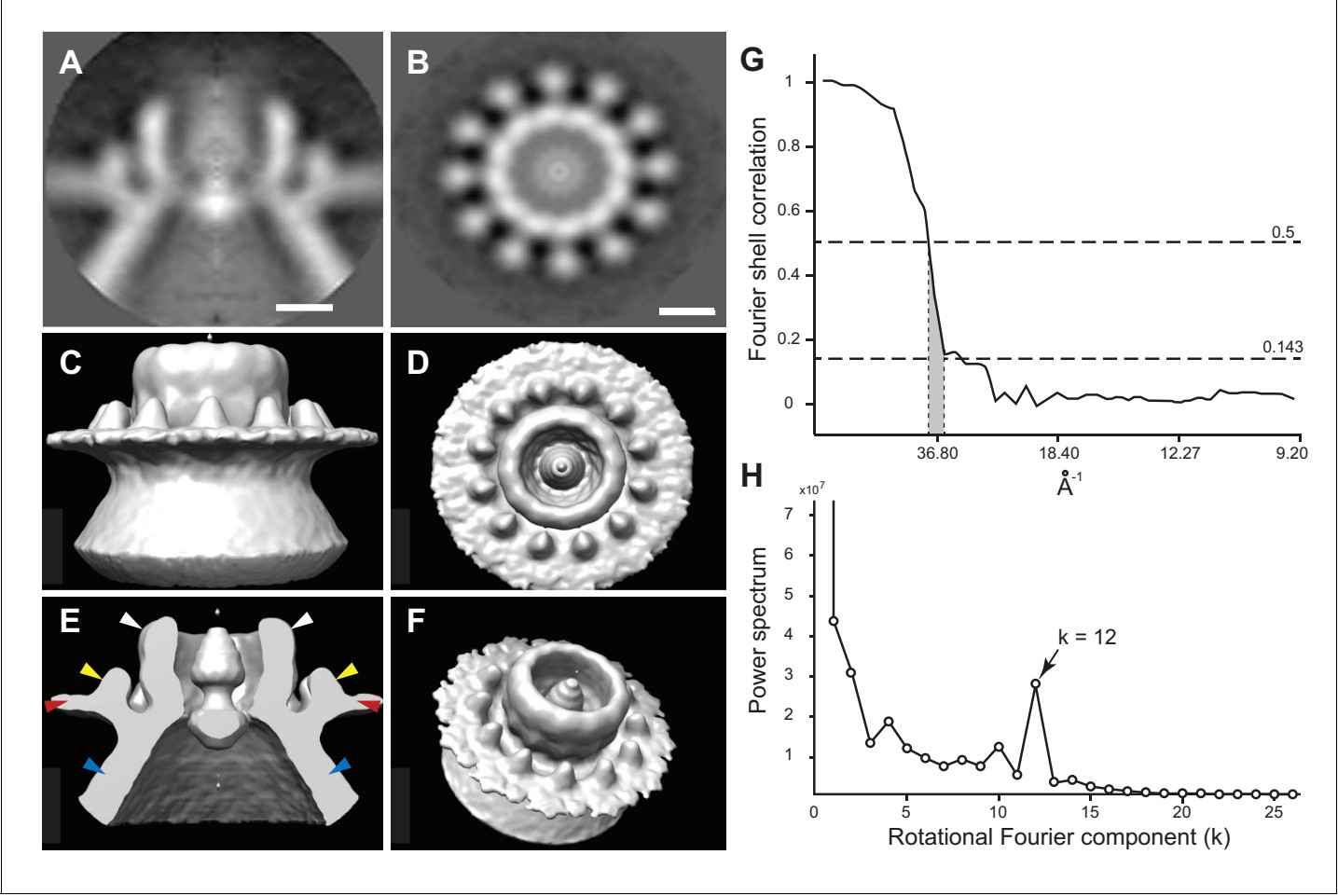

**Figure 6.** Subtomogram average of FHV crown structure. (A–B) FHV crown subtomogram averages viewed from the X/Y plane (A), or from the Z-plane looking towards the spherule interior (B). Scale bars = 10 nm. (C–F) Density maps of FHV crown structures. (C) View of crown density map from the X/Y plane or as depicted in panel A. (D) View of FHV crown from the Z-plane towards the spherule interior as depicted in panel B. (E) Cut-away view of FHV crown. Arrowheads indicate the central turret (white) with interior cone-shaped central density, the teeth surrounding the central turret (yellow), the outer mitochondrial membrane (red), and the spherule membrane (blue). (F) The crown density map angled to show the entirety of all crown features. (G) Fourier shell correlation curve indicating the commonly used 0.5 and 0.143 cut-off intersections indicating the resolution of the reconstruction to be ~3.3–3.8 nm. (H) Averaged power spectra derived from Fourier transformed intensity profiles along the angular direction of all individually analyzed crown particles, strongly indicating 12-fold symmetry.

The following figure supplement is available for figure 6:

**Figure supplement 1.** The 12-fold-symmetric FHV crown.

segment (*Miller and Ahlquist, 2002*; *Miller et al., 2003*), self-interacts through multiple domains in ways critical for RNA replication (*Dye et al., 2005b*), and induces spherule formation in a process strongly linked to copying FHV RNA templates (*Kopek et al., 2010*). Moreover, in the absence of a replication competent FHV RNA template, protein A still localizes to mitochondrial membranes, but 'zippers' adjacent mitochondria together through large patches of regularly spaced electron dense connections, likely reflecting protein A's mitochondrial surface location and strong ability to self-interact (*Dye et al., 2005a*).

To test for the possible presence of protein A in crowns, we used a rabbit antiserum against protein A and gold-conjugated secondary antibodies under cryo-EM conditions. This treatment produced abundant gold labeling on the cytoplasmic side of outer mitochondrial membranes, but only above membrane regions where spherules were present, and not above areas where spherules were

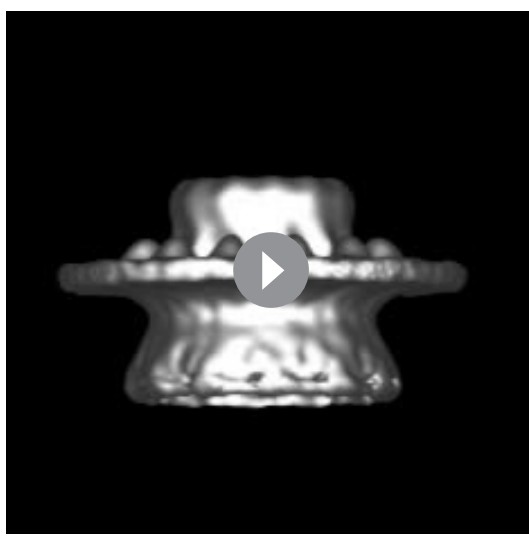

**Video 4.** Subtomogram-averaged FHV crown structure. Rotating views of the highly symmetrical subtomogram-averaged structure at the FHV spherule aperture.

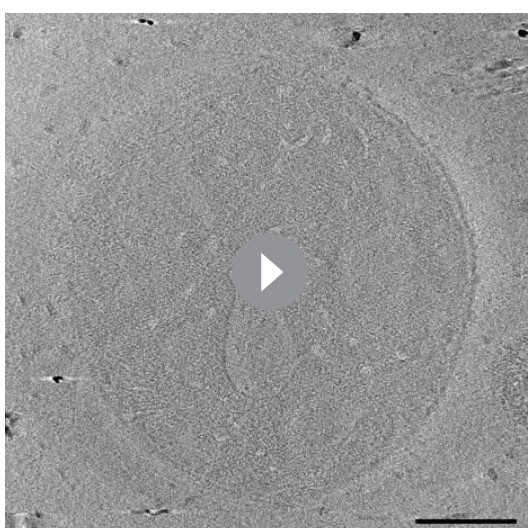

**Video 5.** Cryo-EM tomographic view of a mitochondrion from mock-infected cells. Tomogram of mock-infected mitochondrion as a Z-stack series. In the absence of FHV infection, mitochondria isolated from 2 cells contain no apparent spherules or crown protein complex densities. The inner and outer mitochondrial membranes are closely appressed with little or no membrane dilation. Scale bar = 100 nm.

absent (*Figure 7* and *Video 7*). The distribution of gold particles relative to the outer membrane surface also was well within the expected maximum of ~30 nm estimated by adding two ~ 8 nm primary and secondary antibody lengths and the ~14 nm height of the FHV crown above the membrane surface (see prior section, [*Amiry-Moghaddam and Ottersen, 2013*]). Negative controls with regular rabbit IgG antibodies did not show significant immuno-gold labeling, confirming the specificity of the anti-FHV protein A antiserum (*Figure 7—figure supplement 1*). Also, the high density of anti-protein A gold particles in the vicinity of each spherule (*Video 6*) was potentially consistent with the 12-fold symmetry of crowns (*Figure 6*). Thus, the dense clustering of protein A-targeted immunogold labeling strictly above spherules strongly indicates that protein A is a major component of the FHV crown structure.

## Discussion

Electron microscopy and electron tomography have been used to visualize the basic membrane architecture of RNA replication compartments for a number of positive-strand RNA viruses (*Belov et al., 2007*, *2012*; *Belov and Sztul, 2014*; *Egger et al., 2002*; *Grimley et al., 1968*; *Hatta et al., 1973*; *Knoops et al., 2012*; *Motoyoshi et al., 1973*; *Romero-Brey et al., 2012*; *Welsch et al., 2009*), although for many viruses the exact sites of viral RNA synthesis remain to be determined. Here we report detailed analysis of synthetically active nodavirus RNA replication compartments or 'spherules' by cryo-electron tomography, whose substantial advantages include rapid cryo-fixation of samples in a much more native state and direct imaging of intrinsic electron density without the obscuring effects of heavy metal stains. Among other striking features, the resulting spherule structures and complementary assays reveal filamentous internal RNA templates, a dramatic multimeric protein crown gating the necked connection to the cytoplasm, associated cytoplasmic filaments that likely represent nascent product RNAs, and strong correlation between RNA template length and spherule size. Moreover, as outlined below, these findings motivate a possible model for RNA replication compartment formation and function. Since varied other positive-strand RNA viruses also invaginate intracellular membranes to form comparable spherule RNA replication compartments, many of these results may have implications for other viruses.

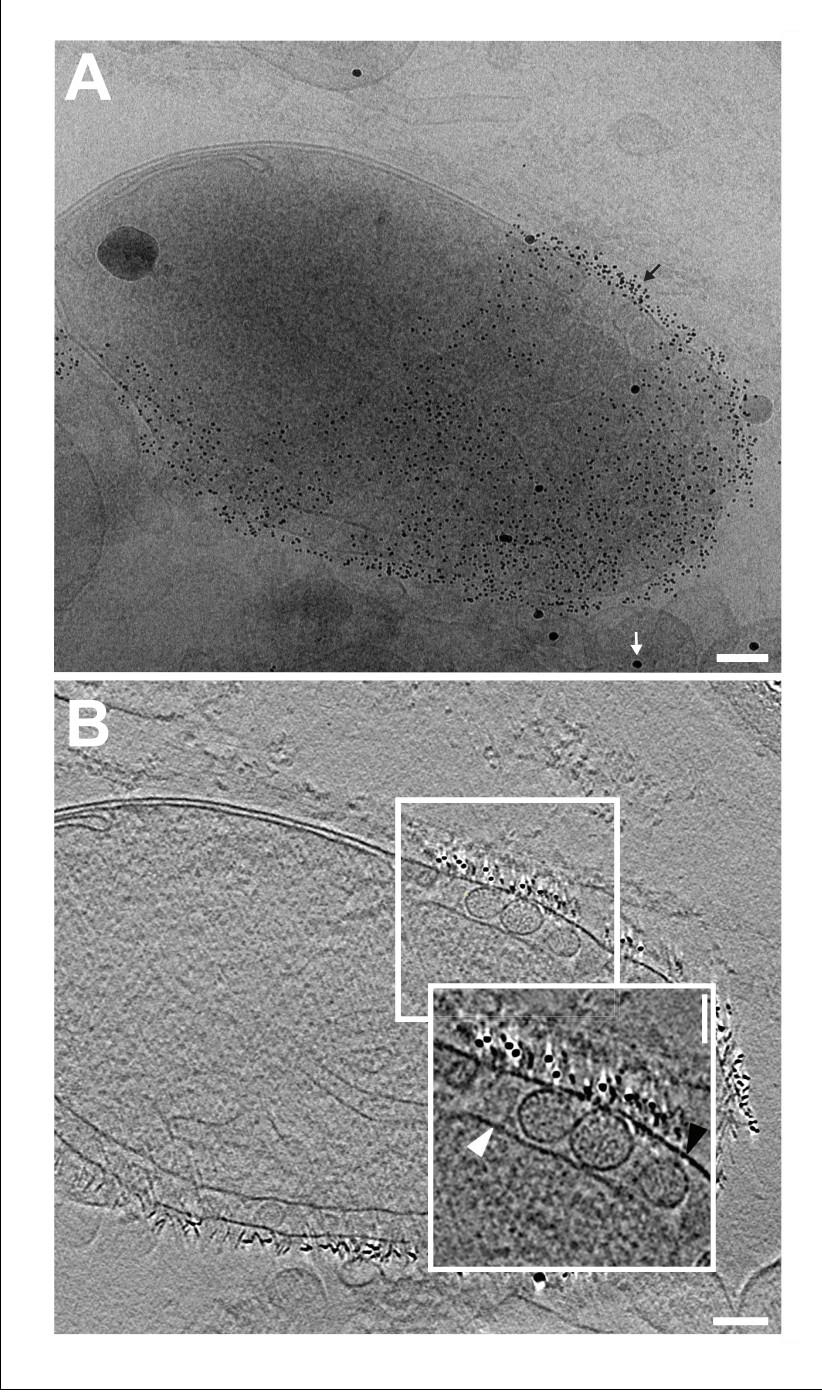

**Figure 7.** FHV protein A replicase is a major component of spherule crown structures. Immunogold labeling of a mitochondrion from FHV-infected cells. (**A**) Mitochondrial extracts were incubated with anti-protein A polyclonal serum and a secondary antibody conjugated to 6 nm gold, loaded onto grids, plunge-frozen, and imaged under cryogenic conditions. White arrows indicate 6 nm gold conjugated to a goat-anti rabbit antibody, black arrows indicate 15 nm fiducial gold used to align the tilt series. (**B**) Image from a reconstructed tomogram of the mitochondrion in panel A. Gold-conjugated antibodies accumulate at the outer mitochondria membrane at sites with spherules, and not in areas where spherules are absent. Inset provides a magnified view and indicate the inner (white arrowhead) and outer (black arrowhead) mitochondrial membranes. Scale bar (**A–B**) = 100 nm, inset scale bar = 50 nm. See *Video 6* for complete tilt series.

The following figure supplement is available for figure 7:

*Figure 7 continued*

**Figure supplement 1.** Lack of immune-gold labeling of FHV spherules using a non-specific antibody control.

## Relationships among spherule shape, filamentous contents, size distributions, and RNA templates

When imaged previously by EM and electron tomography of chemically fixed samples, spherules appeared elongated parallel to the neck axis and with indistinct contents, often shrunken, possibly due to chemical crosslinking (*Kopek et al., 2007*, *2010*; *Miller et al., 2001*). More native preservation by rapid cryo-fixation uniformly revealed nearly round spherules that were densely packed with internal filaments (*Figure 3*). In keeping with numerous results showing that spherules of FHV and many other positive-strand RNA viruses contain positive- and negative-strand template RNAs, likely in dsRNA form (*Hatta and Francki, 1981*; *Kopek et al., 2007*; *Lee et al., 1994*; *Russo et al., 1983*; *Schwartz et al., 2002*), these filaments strongly resemble genomic dsRNAs in reovirus virions (*Zhang et al., 2015*) and dsDNA in DNA bacteriophage (*Cerritelli et al., 1997*; *Liu and Cheng, 2015*).

The average spherule diameter of ~50 nm measured here was similar to earlier estimates (*Kopek et al., 2007*, *2010*; *Miller et al., 2001*). However, prior EM studies using chemical fixation revealed a limited range of FHV spherule sizes and failed to discriminate the size ranges of spherules produced by sgRNA3 replication from spherules produced by full FHV infection (*Kopek et al., 2007*, *2010*; *Miller et al., 2001*). In the current studies, cryo-fixation preserved a much wider, multiple-peaked distribution of spherule sizes from ~14,000 to~145,000 nm$^3$ and linked the replication of sgRNA3 (0.4 kb) and genomic RNA1 (3.1 kb) with the smallest and largest subsets, respectively (*Figure 4*). While there was a clear general correlation between the RNA template length and spherule size, the spherules associated with individual RNA templates showed a range of sizes. Thus, calculations above estimated that

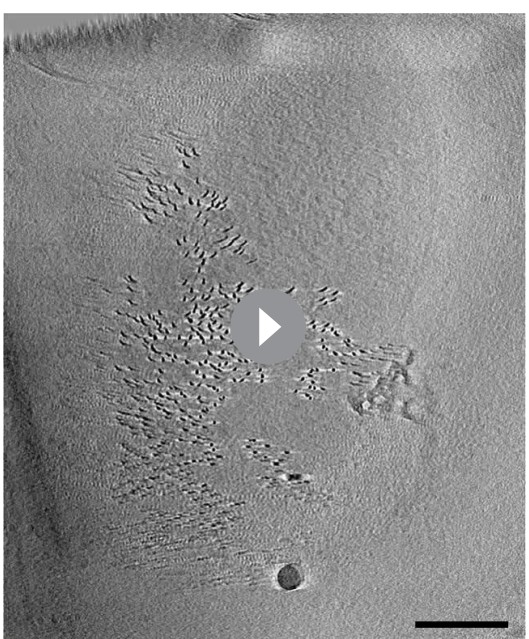

**Video 6.** Immunogold-labeled mitochondrion from FHV-infected cells. Tomogram of the mitochondrion from *Figure 7* represented as a Z-stack series. Note the high accumulation of gold-conjugated anti-FHV protein A antibodies that occurs just outside of the outer mitochondrial membrane in the immediate vicinity of the FHV spherule necks, but is absent from portions of the outer membrane lacking spherules. Scale bar = 200 nm.

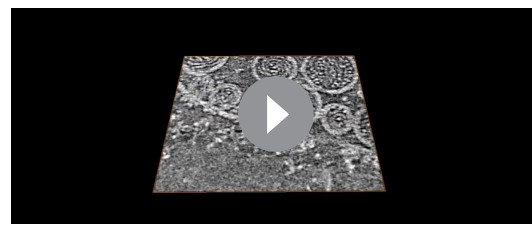

**Video 7.** Three-dimensional image segmentation illustrating multiple features of an FHV spherule. The outer spherule membrane (white) is contiguous with the outer mitochondrial membrane (dark blue). In the first half of the video, a section of the spherule membrane and the underlying spherule interior are purposely removed to show the internal fibrils (red). On the outer mitochondrial membrane at its junction with the invaginated spherule membrane, the crown (light blue) is shown facing toward the cytoplasmic space and surmounting the membrane neck connecting the spherule interior to the extra-mitochondrial cytoplasmic space. The crown density shown is derived from the subtomogram-averaged crown structure of *Figure 6* and *Video 4*. In the final views, the spherule membrane is removed from the image to reveal more detail of the inner fibrillar contents. Segmentation was carried out using the Amira software package (FEI).

the ~80,000–130,000 $nm^3$ spherules associated with RNA1 replication (*Figure 4B*) were large enough to potentially accommodate ~1–3 copies of dsRNA1, while the peak of ~7,000–15,000 $nm^3$ spherules linked with RNA3 replication each might accommodate 1 to 2 copies of dsRNA3 (*Figure 4B and C*). These estimates are consistent with several non-exclusive possibilities, including that some spherules may contain more than one dsRNA, that more than one positive-strand RNA may be simultaneously synthesized on a single negative-strand template, or that export of progeny positive-strand RNAs may not be coincident with their synthesis. In keeping with these possibilities, the sizes of individual spherules might vary over time through RNA synthesis, export, or both. Since the cryo-EM snapshots of these presumably unsynchronized processes likely would not capture all spherules in identical states, such dynamic variation might contribute significantly to the range of spherule sizes associated with specific template RNAs. Future studies are needed to address these important mechanistic questions.

The correlation between spherule volume and RNA template size might be a more general property of positive-strand RNA virus replication. For example, the alphavirus Semliki forest virus (SFV), which replicates its genome in similar spherular replication compartments on the plasma membrane, also shows a correlation of spherule size and template length (*Kallio et al., 2013*).

## Multimeric FHV protein A crowns the cytoplasmic side of spherule apertures

Another unprecedented feature of these replication compartments revealed by cryo-EM was the striking crown structure on the cytoplasmic side of the spherule aperture. Elements visible in individual images and refined by sub-tomogram averaging of hundreds of independent crowns included an ~18 nm high x 19 nm diameter, 12-fold symmetrical, cup-like main body, a central density within the cup, and an outer ring of 12 smaller projections (*Figures 5* and *6* and *Figure 6—figure supplement 1*). Immunogold labeling with antiserum against protein A, which was closely restricted to mitochondrial membrane regions harboring spherules, revealed that crowns contained a high concentration of the highly multifunctional FHV RNA replication protein A (*Figure 7* and *Video 6*). Consistent with the symmetry and dense structure of the crown, our prior fluorescence resonance energy transfer and co-immunoprecipitation experiments show that protein A multimerizes through multiple non-overlapping segments (*Dye et al., 2005a*). Moreover, calculations based on the average density of globular proteins show that the crown volume is sufficient to encompass 12 copies of protein A. However, the present lack of structural information on protein A does not permit meaningful modeling of protein A into the crown. In addition, while these findings show that protein A is a major component of the crown, present results do not rule out that other viral or host proteins might also be recruited into the crown along with protein A.

The crown's prominent position ringing the necked spherule aperture, its extensive engagement with the mitochondrial outer membrane, its 12-fold symmetry, and its dense labeling by antibodies against trans-membrane FHV protein A all imply that the crown plays a major role in stabilizing the high-energy spherule membrane deformation. Interestingly, another prominent example of 12-mer viral protein rings are the DNA packaging portal complexes of herpes simplex virus (*Trus et al., 2004*), bacteriophages Φ29 (*Simpson et al., 2000*), and SPP1 (*Lhuillier et al., 2009*). Similarly, dsRNA *Cystoviridae* encode a hexameric portal protein ring that also translocates parental RNA genome templates into virions and releases nascent viral RNA products to the cytoplasm (*Kainov et al., 2004*). The next section considers multiple findings suggesting that the crowns revealed here might have similar roles in the generation and function of FHV spherules.

While the immunolabeling of intact mitochondria shown in *Figure 7* documents the high density of protein A in the crown, prior immunogold labeling of sectioned mitochondria using a different anti-protein A antibody revealed that some protein A also resides within spherules (*Kopek et al., 2007*). Localization to more than one site, possibly in alternate multimeric forms (*Dye et al., 2005a*), could be related to protein A's many roles in spherule formation, negative- and positive-strand genomic RNA synthesis, sgRNA3 synthesis, and RNA capping (*Ball, 1995*; *Eckerle et al., 2003*; *Johnson and Ball, 1999*; *Kopek et al., 2010*; *Lindenbach et al., 2002*; *Wu et al., 2014*). Estimates of the number of spherules per cell also suggested a ratio of up to 100 protein A molecules per spherule, raising the possibility that protein A might form a lattice within spherules (*Kopek et al., 2007*). While higher resolution is needed to be definitive, our new cryo-EM analysis did not reveal any ordered, membrane-lining lattice in FHV spherules (*Videos 1–3*). Rather, the present study

suggests that prior analysis of chemically-fixed samples might have underestimated the number of spherules per infected cell, leading to an overestimate of protein A copies per spherule. Alternately, a fraction of protein A might be associated with the RNA filaments of the spherule interior, similar to the association of birnavirus VP3 nucleocapsid protein with the dsRNA genome (*Luque et al., 2009*).

## Model for spherule formation and function

*Figure 8* outlines a dynamic model for FHV spherule formation that integrates multiple findings of this study on spherule structure and the linkage of spherule size and template RNA length, with diverse results including the dependence of spherule formation on viral RNA synthesis (*Kopek et al., 2010*). A similar model was proposed in conjunction with Semliki Forest virus (*Balistreri, 2010*; *Kallio et al., 2013*). *Figure 8A*, a single image from *Video 7*, and *Figure 8B*, a corresponding schematic, integrate the new three-dimensional features of the FHV replication complex spherule – the invaginated outer-mitochondrial membrane, the densely-packed interior fibrils, and the aperture

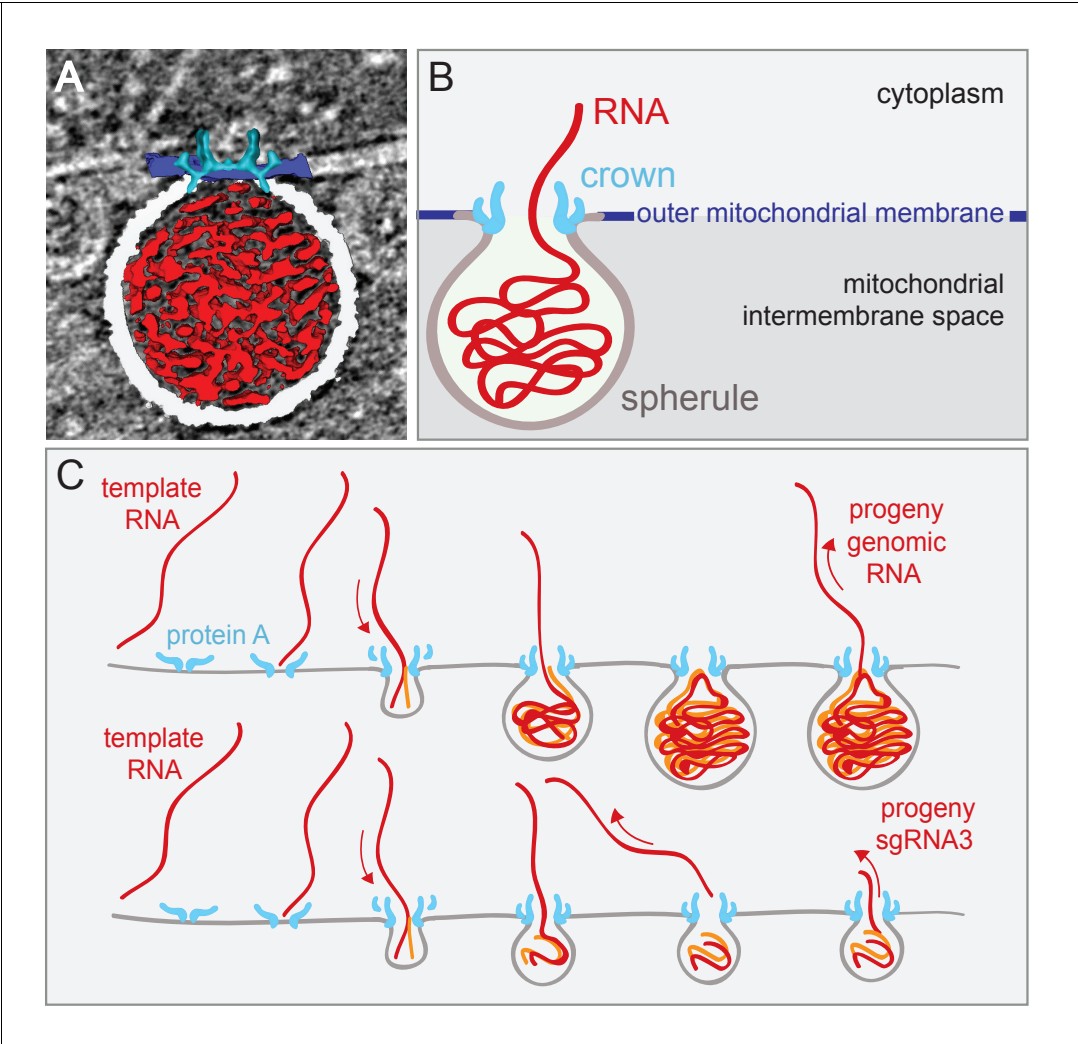

**Figure 8.** Model of FHV replication complex structure and function. (**A**) Amira-assisted 3D segmentation of a single virus replication compartment in FHV-infected cells. The spherule membrane (white) encloses the viral RNA (red) and is anchored to the mitochondrial outer membrane (dark blue), by the FHV protein A crown (light blue, derived from the subtomogram average). (**B**) Cartoon rendering of the image in (**A**). Viral RNA is shown as exiting the spherule to include the observation of fibrillary extrusions in many other tomograms. (**C**) Model of the FHV RNA complex and RNA synthesis, where positive-strand RNA (red) associates with protein A on the mitochondrial membrane to initiate the synthesis of negative-strand RNA (orange) and progeny positive-strand RNA accommodated by increasing volume of the spherule as if blowing up a balloon. In this model, subgenomic RNA3 results from attenuated negative strand synthesis and subse- quent positive-strand production in small spherules.

crown. *Figure 8C* shows this model for FHV RNA replication, in which all steps are strongly based on our prior and new results. As shown previously, protein A localizes to mitochondrial membranes in the absence of other viral factors and, in a step separable from later RNA synthesis and spherule formation, recruits FHV genomic RNA to mitochondrial membranes through specific *cis*-acting RNA signals (*Van Wynsberghe and Ahlquist, 2009*; *Van Wynsberghe et al., 2007*). Subsequent invagination of the outer mitochondrial membrane to form a spherule requires viral RNA synthesis by FHV protein A with an active polymerase domain (*Kopek et al., 2010*). A possible mechanistic unification of FHV RNA synthesis and spherule formation is suggested by the similarity of the 12-fold symmetric, protein A-containing crown to 12-mer genome-loading viral portal protein rings (*Lhuillier et al., 2009*; *Simpson et al., 2000*; *Trus et al., 2004*). Specifically, the dependence on viral RNA synthesis, the densely filament-packed state of spherules (*Videos 1* and *3*), and the scaling of spherule size with RNA template length over a $\geq$10 fold range (*Figure 4*), all would be explained if the spherule was formed by injecting the nascent RNA product of negative-strand RNA synthesis onto the cytoplasmic side on the mitochondrial membrane, causing its deformation and invagination (*Figure 8C*). The FHV crown, strongly stabilized by the trans-membrane domains of multiple copies of protein A, could provide a highly stable platform to drive such genome translocation and the associated lipid flow required to expand the growing spherule vesicle. While virion portal multimers use NTPase activity to translocate viral RNA or DNA, the protein A polymerase domain might directly translocate its initial viral RNA product into the growing vesicle (*Figure 8C*), since single RNA polymerases generate forces of ~25 pN (*Wang et al., 1998*). While substantial, this is half of the maximal force generated by the Phi 29 DNA packaging protein, one of the most powerful biological motors known (*Smith et al., 2001*). Such differences might in part account for the significantly lower RNA packing density in FHV spherules relative to that in reovirus virions, as noted above in the Results.

Regardless of their mechanisms of genesis, established FHV spherules synthesize almost exclusively positive-strand RNAs (*Kopek et al., 2007*), which are capped and released to the cytoplasm (*Figure 8C*). In keeping with their strategic positioning above spherule apertures to the cytoplasm, *Figure 5* and *Videos 1–3* show that crowns are frequently sources of cytoplasmic filaments extending from spherules into the surrounding extra-mitochondrial space. Such RNA release by crowns could be similar to the dsRNA *Cystoviridae* portal protein rings, which actively translocate parental RNA genome templates into virions and passively channel release of nascent positive-strand progeny genomes (*Kainov et al., 2004*). Moreover, in cross-section, the crown is strikingly similar to the turrets of transcriptionally active dsRNA reovirus cores, which both release and add 5' $m^7G$ caps to nascent positive-strand ssRNA products (*Zhang et al., 2015*, *2003*). Since protein A contains a methyltransferase/guanylyltransferase domain involved in capping viral RNAs (*Ahola and Karlin, 2015*; *Johnson and Ball, 1999*), the crown may also be the site of FHV RNA capping,

The calculations above that some spherules may contain more than one dsRNA genome copy imply that either more than one genomic RNA copy can be introduced when the spherule is formed or, as happens for some dsRNA viruses (*Wickner, 1996*), progeny positive-strand RNA may occasionally be retained rather than exported to the cytoplasm. Nevertheless, such progeny positive-strand RNA export must be quite efficient. Late in FHV infection, cells contain ~70–100 copies of positive strand RNA1 or RNA2 per corresponding negative strand genomic RNA (*Kopek et al., 2007*). Thus, even if each mature spherule contained two copies of either FHV genomic RNA1 or RNA2, ~140 to 200 positive-strand RNAs must previously have been synthesized and exported to the cytoplasm per spherule.

In conclusion, cryo-EM tomography has dramatically enhanced our understanding of RNA viral replication complex structure and function. Further advancements in cryo-EM tomography and X-ray crystallography will continue to reveal new and exciting insights into the critical structure-function relationships between viral proteins, viral RNA templates and host membranes that drive positive-strand RNA virus genome replication. Ultimately, such insights should further instruct efforts to interfere with viral replication, including potentially the design of broad-spectrum antivirals.

# Materials and methods

## Cell culture, infection, and transfection

Schneider's *Drosophila* line 2 (S2) cells, purchased from ATCC in the course of these experiments ([D. Mel. (2), SL2] (ATCC CRL1963); RRID:CVCL_Z232), were grown at 28°C in Schneider's medium and 8% fetal bovine serum or in serum-free Express-Five media, supplemented with penicillin, strep-tomycin, and L-glutamine (Gibco). Passages 2–4 of this ATCC stock were frozen and on a regular basis, portions of these early passages were thawed to seed new batches of cells, yielding results that were highly reproducible. Sucrose gradient-purified FHV stocks were produced as previously described (*Friesen and Rueckert, 1984*). *Drosophila* S2 cells were infected at a multiplicity of infection of 10 for all experiments and allowed to proceed for 16 hr unless otherwise noted. DNA plasmid transfections of S2 cells used kit V and G-030 pulse settings on an Amaxa nucleofector (Lonza) or two 30 ms pulses of 1200 V in 100 μL tips on a Neon electroporator (Bio-Rad) and were harvested 48–72 hr post-electroporation.

## Plasmids

Plasmid constructs to express FHV RNA1 (pIE1-FHVRNA1), FHV RNA3 (pIE1-FHVRNA3), or protein A (pIE1PtnA) in *Drosophila* cells under control of the baculovirus immediate-early 1 (IE1) promoter and trans-activating hr5 enhancer have been described previously (*Kopek et al., 2010*). Plasmids to express FHV RNA replication templates additionally contained the hepatitis delta virus anti-genomic ribozyme to ensure precise 3' ends.

## Mitochondrion isolation

Mitochondria were extracted from $6 \times 10^7$ S2 cells using the Q-proteome kit (Qiagen) according to the manufacturer's protocol. Briefly, cells were pelleted and re-suspended in a series of lysis buffers before being physically disrupted by drawing the suspensions ten times through a 25-gauge syringe needle. The resulting cell lysates were fractionated using several rounds of differential centrifugation until a pellet of concentrated mitochondria was obtained.

## FHV RNA synthesis

FHV mitochondria RNA synthesis reactions were assembled using a modification of a previously described protocol (*Barton et al., 1995*). Briefly, re-suspended isolates of mitochondria from FHV-infected cells (45% of total sample volume, with a protein concentration of approximately 30 μg/μL) were incubated in RNA synthesis buffer (final concentration: 1 mM ATP, 0.25 mM CTP and GTP, 0.01 mM UTP, 30 mM creatine phosphate, 20 μg creatine kinase, 20 U RNasin (Promega), 15 mM HEPES (pH 7.4)), 10 μCi [$^{32}$P]UTP (GE Healthcare), to a final volume of 25 μL. Reactions were incu-bated at room temperature and flash-frozen in liquid nitrogen. RNAs were extracted using RNaque-ous spin columns (Thermo-Fisher) according to manufacturer specifications. Following elution, samples were ethanol-precipitated, and RNA pellets were washed with 70% ethanol, dried, and re-suspended in RNase-free water supplemented with formaldehyde loading dye. Samples were elec-trophoresed at 100 mV for 2.5 hr on a 1.0% Tris-borate-EDTA agarose gel. The gel was then dried at 60°C, and exposed to a phosphorimager screen for imaging.

## Immunogold labeling

EM-visualization of protein A replicase on mitochondria used a modification of the protocol from *Yi et al. (2015)*. Briefly, at room temperature, mitochondrial preparations were incubated in PBS supplemented with 5% BSA for 20 min, pelleted at 7000 x g for 2 min and washed three times with incubation buffer (PBS + 0.1% Aurion-BSA(c) (Aurion)). Rabbit polyclonal anti-FHV protein A anti-body (*Miller et al., 2001*) or normal rabbit IgG (Abcam) as a negative control were added at 1:50 dilution and samples were rotated for 1.5 hr. After four washes with incubation buffer, samples were incubated with goat anti-rabbit antibody conjugated to 6 nm colloidal gold (EMS) for 1.5 hr at room temperature. Following four washes, samples were re-suspended in Q-proteome kit storage buffer (Qiagen) supplemented with 10 nm fiducial gold (EMS), and applied to glow-discharged grids for plunge freezing.

## Cryo-electron microscopy

Mitochondrial pellets were re-suspended to a final volume of approximately 50 µL isotonic buffer and supplemented with 10 nm fiducial gold particles (EMS). Copper grids with a 3.5 µm hole-size and 1.0 µm spacing (Quantifoil) were glow-discharged for 30 s using a Pelco GD unit and loaded onto a Mark IV Vitrobot (FEI). 3 µL of the mitochondrial preparation was loaded onto a grid at approximately 95% humidity and blotted for 2 s with Whatman paper at a −2 force offset. Grids were plunge-frozen in liquid nitrogen-cooled liquid ethane and stored under liquid nitrogen prior to loading into the cryo-EM holder and the microscope. Cryo-tomograms were acquired using either a Titan Krios electron microscope at 300 keV (FEI) at Janelia Research Campus (HHMI) with a pixel size of 2.9 Å or 5.6 Å or on a TF-30 electron microscope at 300 keV (FEI) at UW Madison with a pixel size of 4.6 Å. Both microscopes were equipped with post-column energy filters and post-GIF K2 Summit direct electron detectors (Gatan). Low and medium magnification maps were acquired using either UCSF tomo (*Zheng et al., 2007*) or SerialEM (*Mastronarde, 2005*). Tomograms were collected at 2° tilt increments for a range spanning −60° to +60° using defocus values ranging from 4 to 6 µm. The total electron dose per tomogram averaged 130–150 electrons/Å$^2$. The energy filter slit size during collection ranged between 10–25 eV on both microscopes.

## Subtomogram averaging

The FHV crown data set included 6 tomograms, each containing between one and four mitochondria. Mitochondrial membranes were segmented semi-automatically (*Castaño-Díez et al., 2017*), and modeled as surfaces. Particles were located by visual inspection, and each was assigned an initial orientation given by the normal to the membrane. Using the Dynamo software (*Castaño-Díez et al., 2012*), an initial template was created by averaging a total of 320 extracted particles, using the initial orientation. Refinement iterations were computed by scanning orientations in a cone around the initial orientations. The initial search range was 60° around the initial axis, and a full azimuthal rotation range of 360°. Particles were forced to stay in a neighborhood around their original location. Convergence was observed after 12 iterations (with an angular search range of 5° around the axis located at the previous iteration). The resolution of the resulting average was estimated at 3.3 nm (criterion FSC = 0.143, *Figure 6G*), by dividing the total set of particles into two data sets refined independently. Subtomogram averaging was independently achieved by a second researcher using the Particle Estimation for Electron Tomography (PEET) set of programs from the IMOD tomography suite (*Nicastro et al., 2006*; RRID_SCR_003297) without prior knowledge of the results obtained using Dynamo. The independent Dynamo and PEET subtomogram reconstructions revealed essentially identical structures, providing assurance of negligible unintentional error or bias.

## Acknowledgements

For cryo-EM tomography, we thank Jason de la Cruz, Chuan Hong, Rick Huang, and Zhiheng Yu for assistance with the HHMI Janelia Farm Research Campus facilities, Alex Kvit and Jerry Hunter for assistance with the University of Wisconsin – Madison Materials Science Center (MSC) facilities, and Z Hong Zhou for the use of facilities at the University of California – Los Angeles Electron Imaging Center for NanoMachines (EICN) during preliminary stages of the project. We also thank Daniel Toso for assistance with tomogram reconstructions, Lauren Michaels and Miron Livny at the Morgridge Institute for Research for computer resource assistance, and members of our own laboratory for helpful discussions.

## Additional information

### Funding

| Funder | Grant reference number | Author |
|---|---|---|
| Howard Hughes Medical Institute | Investigator | Paul Ahlquist |
| Morgridge Institute for Research | Investigator | Paul Ahlquist |

| National Science Foundation | DBI 1126441 | Marisa S Otegui<br>Paul Ahlquist |
| National Science Foundation | MCB 1614965 | Marisa S Otegui |
| Rowe Family Virology Venture Fund | Investigator | Paul Ahlquist |
| National Institutes of Health | T32 AI078985 | Desirée Benefield |

The funders had no role in study design, data collection and interpretation, or the decision to submit the work for publication.

## Author contributions

KJE, DB, Conceptualization, Investigation, Writing—original draft, Writing—review and editing; DC-D, Software, Investigation, Methodology, Writing—review and editing; JGP, MH, Investigation, Methodology; JAdB, Conceptualization, Supervision, Investigation, Writing—original draft, Writing—review and editing; MSO, Supervision, Funding acquisition, Investigation, Writing—review and editing; PA, Conceptualization, Supervision, Funding acquisition, Investigation, Writing—original draft, Writing—review and editing

## Author ORCIDs

Johan A den Boon, http://orcid.org/0000-0002-5507-917X
Paul Ahlquist, http://orcid.org/0000-0003-4584-9318

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
