## [Decision Letter]

Thank you for submitting your article "Cryo-electron tomography reveals novel features of a viral RNA replication compartment" for consideration by *eLife*. Your article has been favorably evaluated by Wenhui Li (Senior Editor) and two reviewers, one of whom, Wesley I Sundquist (Reviewer #1), is a member of our Board of Reviewing Editors.

The reviewers have discussed the reviews with one another and the Reviewing Editor has drafted this decision to help you prepare a revised submission.

The authors report cryo-EM tomographic reconstructions of Flock house Virus viral replication "spherules" formed as invaginations in the outer mitochondrial membrane. The structure is a relatively low resolution (nominally 32 A) but nevertheless reveals a number of new features that had not been characterized, including a 12-fold symmetric "crown" structure that contains viral replicase protein A, coiled dsRNA in the spherule interior, an unappreciated range of spherule volumes and a correlation between transcript size and spherule volume (which the authors interpret to mean that each spherule contains 1-2 copies of a single specific viral RNA), and what appear to be progeny (positive sense) RNA strands being released from the spherule through the crown.

Despite the relatively modest resolution of the reconstruction, the spherule structures are remarkable and other strengths of the paper include: 1) the experiments are generally well conceived and controlled, 2) the data provide strong support for models for spherule biogenesis and activity (summarized in Figure 7), and 3) the potential generalizability of these structures to the many other positive sense RNA viruses that also build replication factories on cellular membranes.

Significant issues that must be addressed:

1) The paper is generally straightforward and well written, but details of the reconstruction methodology and statistics are sparse (e.g., the FSC curve is not shown, the meaning of the sentence "Local resolution estimate of 3.2 nm" is unclear -what does "local" mean, what is the average resolution of the structure, etc.). Most importantly, the assumption of 12-fold symmetry is not sufficiently justified – for example by providing a power spectrum showing a clear peak corresponding to 12-fold symmetry. There is no reason to think this assumption is incorrect, but the symmetry and structure of the crown are key new features and the experimental underpinnings should therefore be presented in detail.

2) Interpretations of spherule size and content are an important component of this paper, but require clarification. There are several issues:

A) Please explain exactly what is meant by the sentence: "Consistent with estimates of the persistence length of dsRNA" – didn't understand this sentence.

B) Figure 3: Should address whether/how RNA2 explains the remaining size classes of spherules between those assigned to RNA1 and RNA3.

C) The interpretation of Figure 3 is that differing size RNAs and their associated replication results in different size classes of spherules. Is it alternatively possible that spherules are dynamic and not all of them are being captured at the same time during imaging? Therefore, spherules are at different stages of "growth" in this instance but may contain all the RNAs.

D) Please clarify further the calculations between template length and spherule volume were derived and what relationship they have to cytoplasmic polyhydrosis virus (CPV) dsRNA. Perhaps it would be useful to show Figure 2 together with a table to comparing internal volumes of CPV and FHV spherules together.

E) Postulating that 1.5 – 2.5 copies of dsRNA1 fit inside each spherule is not particularly satisfying or intuitive. Since multiple (+) strands come from the (-) strand wouldn't it be more logical to frame this argument in the context of a single (-) strand and multiple nascent (+) strands?

Other issues that should be addressed:

Introduction

• "Understanding and potential antiviral manipulation would be greatly assisted[…]" – clumsy sentence.

• It would be valuable to provide a brief discussion of structural information on Protein A that would be relevant and helpful to readers who are not familiar with Nodavirus research.

Results

• Figure 2) would be useful to use arrows and identify crown, turrets, filaments as in Figure 5). Scale bars would allow a better idea of what magnification and scale is being shown. E) Unclear when referring to Figure 2 in the results as to why distance between interior filaments plot is shown and how it is calculated.

• Subsection “Spherule replication compartment volume correlates with RNA template length”, first sentence is missing a period.

Discussion

• The composition of the crown is not adequately discussed. It is clear that Protein A is involved but does it make up the crown or not? Is it possible that a host protein is co-opted? Are there any predictions about the structure of protein A that would allow fitting the structure into the crown? From the data it appears that the crown is likely to be a motor akin to the phage packaging motors, which was indeed mentioned, but a possible role in assembly was never discussed.

---

## [Author Response]

*Significant issues that must be addressed:*

*1) The paper is generally straightforward and well written, but details of the reconstruction methodology and statistics are sparse (e.g., the FSC curve is not shown, the meaning of the sentence "Local resolution estimate of 3.2 nm" is unclear -what does "local" mean, what is the average resolution of the structure, etc.).*

In the Results subsection “FHV spherule necks are crowned by cupped structures”, second paragraph, we have added further description of the reconstruction methodology and statistics. This added section refers to new panel G in what is now Figure 6 to provide the Fourier Shell Correlation (FSC) curve in support of statements on the structure’s resolution. We agree that use of “local” to describe resolution was unnecessary and confusing, and have omitted that term.

*Most importantly, the assumption of 12-fold symmetry is not sufficiently justified – for example by providing a power spectrum showing a clear peak corresponding to 12-fold symmetry. There is no reason to think this assumption is incorrect, but the symmetry and structure of the crown are key new features and the experimental underpinnings should therefore be presented in detail.*

The power spectrum in question is now shown in new panel H of Figure 6, and is introduced in the new comments in the second paragraph of the subsection “FHV spherule necks are crowned by cupped structures”, as the basis for the statements on crown symmetry. We have included clarifying statements that no symmetry was imposed for subtomographic reconstruction, and that individual tomograms prior to averaging show clear indications of 12-fold symmetry in displaying parts of the outer ring of 12 tooth-like projections.

*2) Interpretations of spherule size and content are an important component of this paper, but require clarification. There are several issues:*

*A) Please explain exactly what is meant by the sentence: "Consistent with estimates of the persistence length of dsRNA" – didn't understand this sentence.*

At the end of the subsection “Coiled interior filaments fill FHV RNA replication vesicles”, we have provided further information and explanation to clarify that the presence of relatively straight, continuous segments of at least 40 nm is consistent with the prior estimate of dsRNA persistence length, i.e. the length of dsRNA required to show significant bending, as ~60- 64 nm.

*B) Figure 3: Should address whether/how RNA2 explains the remaining size classes of spherules between those assigned to RNA1 and RNA3.*

In the relevant Results section, we have added a new paragraph (subsection “Spherule replication compartment volume correlates with RNA template length”, third paragraph) noting that the breadth of spherule sizes between those assignable to RNA1 and RNA3 appears greater than expected for a single peak associated with RNA2. We also note that potential contributions to this complexity could include the dependence of RNA2 replication on trans-activation by RNA3, which could lead to some spherules containing both RNA2 and RNA3, and the possibility, noted in reviewer comment 2C immediately below, that spherule sizes might vary dynamically through ongoing RNA synthesis and export. Since further comments on these points necessarily involve alternatives that cannot be resolved from present data, readers then are referred to the Discussion, where a fuller consideration of factors possibly contributing to the breadth of spherule sizes has been added (subsection “Relationships among spherule shape, filamentous contents, size distributions, and RNA templates”, third paragraph).

*C) The interpretation of Figure 3 is that differing size RNAs and their associated replication results in different size classes of spherules. Is it alternatively possible that spherules are dynamic and not all of them are being captured at the same time during imaging? Therefore, spherules are at different stages of "growth" in this instance but may contain all the RNAs.*

Please see the response to comment 2B immediately above. As indicated there, the possibility that spherule sizes may vary dynamically is now noted in the subsection “Spherule replication compartment volume correlates with RNA template length”, third paragraph and discussed further in the subsection “Relationships among spherule shape, filamentous contents, size distributions, and RNA templates”, third paragraph.

*D) Please clarify further the calculations between template length and spherule volume were derived and what relationship they have to cytoplasmic polyhydrosis virus (CPV) dsRNA. Perhaps it would be useful to show Figure 2 together with a table to comparing internal volumes of CPV and FHV spherules together.*

In the last paragraph of the subsection “Spherule replication compartment volume correlates with RNA template length”, we have more explicitly explained that we used published measurements of the diameter and packaging density of cytoplasmic polyhydrosis virus (CPV) dsRNA in dsRNA virus virion particles as a reference to calculate packaging density of FHV dsRNA in spherules. We appreciate the suggestion of a table comparing the CPV and FHV compartment volumes, but after testing some alternatives concluded that the presentation was simpler and more straightforward without this addition.

*E) Postulating that 1.5 – 2.5 copies of dsRNA1 fit inside each spherule is not particularly satisfying or intuitive. Since multiple (+) strands come from the (-) strand wouldn't it be more logical to frame this argument in the context of a single (-) strand and multiple nascent (+) strands?*

We agree that discussing spherule volumes in units of fractional dsRNA equivalents is not mechanistically well-founded and would only confuse readers. The relevant section in the last paragraph of the subsection “Spherule replication compartment volume correlates with RNA template length”, has been replaced with more meaningful comments noting that (i) these calculations show that even the smallest spherules associated with RNA1 and RNA3 (Figure 4) are large enough to contain a single dsRNA1 or dsRNA3 template, respectively, and (ii) multiple mechanisms contributing to larger spherule volumes are considered in the Discussion. The corresponding Discussion section (subsection “Relationships among spherule shape, filamentous contents, size distributions, and RNA templates”, second paragraph) includes the potential for synthesis of multiple positive strands from a single negative strand, as well as the potentially dynamic nature of spherule volumes and other points raised in comments 2B and 2C above.

*Other issues that should be addressed:*

*Introduction*

• "Understanding and potential antiviral manipulation would be greatly assisted[…]" – clumsy sentence.

The sentence in question was not essential and was simply deleted.

*• It would be valuable to provide a brief discussion of structural information on Protein A that would be relevant and helpful to readers who are not familiar with Nodavirus research.*

To provide such introduction to protein A and nodavirus biology, we added a new Figure 1 illustrating FHV genome organization (panel A) and a map of summarizing current knowledge on Protein A’s structure and function (panel B). This new figure and protein A structure/function in particular are discussed in the third paragraph of the Introduction.

*Results*

• Figure 2) would be useful to use arrows and identify crown, turrets, filaments as in Figure 5.

As requested, we have added arrowheads in what is now Figure 3, and also in what is now Figure 5, to indicate spherule membrane, crowns and internal filaments. For consistency, the same arrowhead color scheme from what is now Figure 6 (old Figure 5) is used in new Figure 3 and 5.

*B – D). Scale bars would allow a better idea of what magnification and scale is being shown.*

As suggested, we have added scale bars to what is now Figure 3, panels B-D.

*E) Unclear when referring to Figure 2 in the results as to why distance between interior filaments plot is shown and how it is calculated.*

The plot showing the distribution of inter-filament distance in what was Figure 2 was indeed not discussed in the text until a later section on calculating RNA packaging density. To avoid this confusion, this plot was removed from that early figure and is now provided as a Supplement to what is now Figure 3, and cited at the appropriate point in the Results (subsection “Spherule replication compartment volume correlates with RNA template length”, last paragraph). The figure supplement legend now notes how the distances were measured and the aforementioned paragraph explains how these data were used to calculate RNA packing density inside the spherules.

*• Subsection “Spherule replication compartment volume correlates with RNA template length”, first sentence is missing a period.*

We have added the missing period.

*Discussion*

*• The composition of the crown is not adequately discussed. It is clear that Protein A is involved but does it make up the crown or not? Is it possible that a host protein is co-opted? Are there any predictions about the structure of protein A that would allow fitting the structure into the crown?*

While the composition of the crown is a question of great importance, no further information is available at present. To address this, in the first paragraph of the subsection “Multimeric FHV protein A crowns the cytoplasmic side of spherule apertures”, we have further clarified how the immunogold labeling and additional results are consistent with protein A being a major component of the crown. We also added sentences noting that while protein volume calculations show that crowns are sufficiently large to accommodate 12 protein A molecules, no structural information is available for protein A to a level that would permit meaningful modeling into the crown, and that not only host proteins but additional viral proteins might be recruited into the crown with protein A.

*From the data it appears that the crown is likely to be a motor akin to the phage packaging motors, which was indeed mentioned, but a possible role in assembly was never discussed.*

The potential molecular motor-like role of the FHV crowns in driving RNA translocation into growing spherules, via the force of RNA polymerization, is discussed in the context of the proposed model for FHV replication complex assembly in the first paragraph of the subsection “New model for spherule formation and function”.